# Assessment of Cu(II) Removal from Aqueous Solutions by Modified Pomelo Peels: Experiments and Modelling

**DOI:** 10.3390/molecules28083438

**Published:** 2023-04-13

**Authors:** Ruixue Zhang, Mengqing Jiao, Nan Zhao, Johan Jacquemin, Yinqin Zhang, Honglai Liu

**Affiliations:** 1Hebei Province Key Laboratory of Sustained Utilization & Development of Water Resources, Hebei GEO University, Shijiazhuang 050031, China; 2School of Chemistry and Molecular Engineering, East China University of Science and Technology, Shanghai 200237, China; 3Materials Science and Nano-Engineering MSN Department, Mohammed VI Polytechnic University, Lot 660-Hay Moulay Rachid, Ben Guerir 43150, Morocco; 4School of Water Conservancy and Hydroelectric Power, Hebei University of Engineering, Handan 056038, China

**Keywords:** agricultural wastes bio-sorbents, wastewater treatment, heavy metal adsorption, artificial neural network, adsorption mechanism

## Abstract

In this study, low-cost pomelo peel wastes were used as a bio-sorbent to remove copper ions (e.g., Cu(II)) from aqueous solutions. Prior to testing its Cu(II) removal capability, the structural, physical and chemical characteristics of the sorbent were examined by scanning electron microscope (SEM), Fourier transform infrared (FTIR) spectroscopy, and Brunauer–Emmett–Teller (BET) surface area analysis. The impacts of the initial pH, temperature, contact time and Cu(II) feed concentration on the Cu(II) biosorption using modified pomelo peels were then assessed. Thermodynamic parameters associated to the biosorption clearly demonstrate that this biosorption is thermodynamically feasible, endothermic, spontaneous and entropy driven. Furthermore, adsorption kinetic data were found to fit very well with the pseudo-second order kinetics equation, highlighting that this process is driven by a chemical adsorption. Finally, an artificial neural network with a 4:9:1 structure was then established for describing the Cu(II) adsorption using modified pomelo peels with R^2^ values close to 0.9999 and to 0.9988 for the training and testing sets, respectively. The results present a big potential use of the as-prepared bio-sorbent for the removal of Cu(II), as well as an efficient green technology for ecological and environmental sustainability.

## 1. Introduction

Copper is a very toxic metal found in wastewater streams, as copper is commonly used in various industries including electroplating, pigment, iron and steel production, mining and metal finishing [1]. However, any excess of copper intake in the human body causes direct impacts such as hair loss, headache, central nervous system disorder and gastrointestinal dysfunction [2]. In fact, removing copper ions from industrial wastes before releasing them into aquatic habitats is essential. The conventional processes used to date for removing the metals from wastewater are based on chemical precipitation, electrolysis, chemical oxidation and reduction, membrane filtration, ions exchange, etc. [3]. Nevertheless, most of them involve high capital costs and sludge disposal problems [4]. To date, in order to develop sustainable processes, several low-cost materials, used as adsorbents, have been investigated to remove heavy metals from industrial wastewater.

Agricultural wastes present several advantages to be used as adsorbents as they are abundant, cheap and often renewable [5]. However, due to the absence of favorable active sites and poor mechanical strength that might boost the targeted adsorptions, their heavy metals adsorption capacity is often limited [6]. However, specific chemical modifications of the agricultural wastes’ structure could counterbalance this drawback by increasing their potential for heavy metals adsorption [7]. For example, such an adsorbent could be easily oxidized with a chemical treatment using an acid or a base by forming functional groups onto the material surface. The presence of such functional groups onto the surface enhances the material ability to interact with heavy metals, boosting thus its extraction capacity [8]. Other methods, such as the graft polymerization that introduces long polymer chains, could be also used to finetune the agricultural wastes’ structure prior to be reused as bio-sorbents for heavy metals removal [9].

Pomelo peel, a kind of agricultural waste that predominantly constitutes cellulose, pectin, lignin and hemicellulose exhibits a very high biosorption potential. These peels have phenolic acid groups and carboxylic moieties. All of these chemical functions could be implicated in the binding of metals. Recently, several reports highlighted the use of pomelo peels to remove pollutants, like Cr(III), Pb(II), Cd(II), carbamazepine, tetracycline antibiotics, etc. from water streams [10,11,12,13]. However, no information has been reported on utilizing pomelo peels for Cu(II) adsorption in aqueous solutions, to date. Additionally, as many process parameters associated to the adsorption mechanism result in complicated nonlinear relationships, the simulation of any biosorption process is very challenging. In fact, it is difficult to mimic the gathered batch biosorption findings by using a single traditional mathematical model. Artificial neural networks (ANN) offer an alternate method for modeling heavy metal biosorption due to their potential to explain complicated and nonlinear connections between variables [5,14,15,16].

Hence, this study recommends a novel chemically modified pomelo peels bio-sorbent for extracting copper ions present in aqueous solutions. The capacity of the tested material to bio-sorb Cu(II) was investigated by probing the impacts of various operational variables, such as temperature, contact time, initial Cu(II) concentration and initial pH. More importantly, Cu(II) adsorption mechanism using modified pomelo peels was further studied by analyzing the batch experimental data using isothermal, kinetic and thermodynamic models along with an analysis of the materials structure (SEM, BET and FTIR). This analysis was done to provide insights into simulating the associated biosorption process. In addition, a three-layer ANN structure was then established to further evaluate the Cu(II) removal capacity of the tested material. Lastly, to further assess this ANN model’s predictive capacity, model outputs were discussed and compared with all collected experimental data. Therefore, the objective of the present paper was to develop from agricultural wastes a highly efficient bio-sorbent for the removal of Cu(II) from aqueous solutions to reduce the adverse effects on the environment and produce fresh effluents.

## 2. Results and Discussion

### 2.1. Effect of the Chemical Modification on Adsorption

Approximately 56% of the dry weight of the pomelo peels is made up of lignin and cellulose. In fact, hemicellulose is readily hydrolyzed by dilute acid or base. Unlike hemicellulose, cellulose and lignin are indeed robust and resistant to hydrolysis. A chemical modification of the as-received pomelo peels was firstly done, as this material is mainly based on methyl ester groups interacting weakly with metals. In fact, this moiety is not responsible for metal binding in bio-sorbent applications. In contrary, carboxyl groups (-COOH) do show a stronger ability to bind metal ions. Thus, in this work, methyl esters were transformed into carboxylate ligands through handling the peels with NaOH solution. However, pectin, as one of the main components of pomelo peels, contains methyl ester groups. Pectate generated by addition of NaOH is soluble in the aqueous solution, leading, thus, to a reduction in number of carboxyl groups exposed on the adsorbent surface. Furthermore, CaCl_2_ was then added to precipitate pectin due to the formation of Ca-polygalacturonate gels and retain pectin in the adsorbent materials [17]. Therefore, by following the as-proposed chemical modification on the pomelo peels, one can expect an increase of the number of carboxylate ligands on the sorbent structure and, in fact, an increase of its metal-binding ability. Thus, the Cu(II) adsorption capacity data of the pomelo peels before and after this chemical treatment were then compared to highlight their binding abilities. As expected, an enhancement of Cu(II) adsorption capacity within the range of 3.38–4.72 mg/g could be observed after this chemical modification by treating a 20 mg/L of Cu(II) solution at the initial pH = 5.0 with 0.1 g of each adsorbent at 25 °C.

### 2.2. Adsorbent Characterization

The surface features of the pomelo peels before and after the proposed chemical modification were investigated by SEM. Figure 1 displays the images of the pomelo peels (PP) and the modified pomelo peels (MPP) obtained by SEM. As displayed in this figure, the surface morphology of MPP varies greatly from that of PP. In fact, PP has a loose and lamella structure, while MPP surface is more uneven.

Therefore, one can imagine thanks to Figure 1 that the specific surface area of MPP is higher than that of PP. This was further evaluated by investigating the pore size distribution in both PP and MPP materials using a N_2_-adsorption technique as shown in Figure 2a. By looking at the pore size distribution of the pomelo peels, one can clearly notice that few micropores (<2 nm) are present in both structures. However, as displayed in Table 1, the chemical modification of pomelo peels induces structural changes highlighted by an increase of the BET surface area, average pore diameter and pore volume when comparing MPP data with those collected for PP. In fact, both the improvement in pore volume and specific surface area could be related to the enhancement of the Cu(II) biosorption capacity of the tested material after applying the proposed chemical treatment.

Figure 2b displays FTIR spectra of PP and MPP. The wide and intense adsorption peaks at 3333 cm^−1^ for PP and 3336 cm^−1^ for MPP link to O-H stretching vibrations occurring in cellulose, pectin, absorbed water, hemicellulose, and lignin. The peaks located at 2927 cm^−1^ for PP and at 2922 cm^−1^ for MPP can be ascribed to C-H stretching vibrations of methyl, methylene and methoxy groups. Furthermore, the appearance of peaks at 1611 cm^−1^ and 1748 cm^−1^ indicates the stretching bands of carboxylate ion (COO^−^) and ester carbonyl groups (C=O), respectively. The vibrations at 1436 cm^−1^ for PP and at 1429 cm^−1^ for MPP can be associated with the existence of aromatic and aliphatic groups (C-H) corresponding to the plane deformation vibrations occurring in methyl, methylene and methoxy groups. The bands between 1000 cm^−1^ and 1300 cm^−1^ can be attributed to C-O stretching vibrations of alcohols and carboxylic acids. By comparing the FTIR spectra, one can see the peak at 1748 cm^−1^ disappears from the MPP spectra indicating that the methyl ester moieties were hydrolyzed by using NaOH and then transformed into COO^−^.

Based on its porous structure, uneven surface and a higher capacity of free carboxyl groups, one could draw a conclusion that MPP presents interesting physical and chemical features more compatible with the metal bindings than the as-received PP.

### 2.3. Effect of Initial pH

The influence of the initial pH of tested Cu(II) solutions on the material adsorption capacity was investigated over a pH range from 2.0 to 7.0. Prior the adsorption, the initial pH of Cu(II) solution was adjusted to the desired value to avoid the addition of ions during the adsorption measurements. In fact, the pH was not thus maintained constant during the Cu(II) adsorption measurements to avoid additional competition between H_3_O^+^ and Cu^2+^ for active binding sites on the adsorbent surface as recommended by Benaïssa et al. [18]. During these experiments, 0.1 g of MPP adsorbent was used along with 25 mL of a given 20 mg/L tested Cu(II) solution at 25 °C as a function of its initial pH. The percentage of adsorbed Cu(II) (R(%)) was then calculated using data collected after 1 h.

As shown in Figure 3a, the Cu(II) adsorption capacity of MPP shows a minimum at pH = 2.0, which could be attributed to the high concentration of H_3_O^+^ in solution, and by their high mobility. Indeed, at this low pH or even lower, the low value of R(%) reveals that H_3_O^+^ ions seem to compete with Cu(II) ions, i.e., both species seem to interact with the same adsorption sites. Meanwhile, the presence of H_3_O^+^ further induce the presence of the non-ionic carboxyl group, e.g., -COOH onto MPP, resulting to an increase of the Cu(II) electrostatic repulsion and, thus, to a low value of R(%). As expected, the Cu(II) adsorption capacity of MPP then increases by increasing the pH value from 2.0 to 3.0. A plateau of R(%) values is then reached at around pH = 3.0 to 5.0. In other words, pH has a significant impact on the speciation of the tested metal and the active binding sites on the adsorbent surface. At pH > 5.0, the R(%) values decrease slightly due to the formation of copper hydroxide in the solution. At pH = 6, copper hydroxide significantly dominates the species in solution [19]. As a consequence of these measurements, the optimum pH for the Cu(II) adsorption using MPP was set at pH = 5.0 and all further adsorption experiments were thus carried out at this pH.

### 2.4. Adsorption Kinetics

The impact of the contact time on the Cu(II) adsorption of MPP using 25 mL of an initial 20 mg/L Cu(II) solution at the initial pH = 5.0 at 25 °C is depicted in Figure 3b. This plot demonstrates that the Cu(II) adsorption rate of MPP rises very quickly to reach a value of 4.15 mg Cu(II)/g MPP within the first 10 min, indicating a fast kinetic. After a very efficient and rapid Cu(II) adsorption, the rate slowly decreases, indicating that the equilibrium is reached after 60 min. Initially, the active adsorption sites of MPP are free and Cu^2+^ ions bind easily to these sites. In addition, the driving force caused by the concentration gradient between the solid-liquid interface and the bulk solution is strong. Therefore, a rapid adsorption rate is obtained. This is indeed requested for decreasing both the reactor volume and reaction time [20]. After this initial period, e.g., herein the first 10 min, a slower solute diffusion into this adsorbent structure induces a decrease of the adsorption rate until reaching the adsorption capacity of MPP. Therefore, in the light of this kinetics experiment, a 60-min of contact time was eventually selected as the adsorption time to ensure reaching the equilibrium.

The kinetic of the Cu(II) adsorption of MPP was then further analyzed by using the following pseudo-second order equation:(1)$$\frac{t}{q_{t}}=\frac{1}{k_{2}q_{e}^{2}}+\frac{t}{q_{e}}$$
where *k*_2_ represents the constant of pseudo-second order rate; *q*_e_ and *q*_t_ denotes the adsorption capacity at equilibrium and the adsorption capacity at time *t*, respectively.

As shown in Figure 4, an excellent linear fitting between the inverse equilibrium adsorption capacity and the contact time is observed, resulting that Equation (1) could be utilized in fitting collected data at 25 °C. In fact, the *q*_e,exp_ value determined thanks to the experimental data is close to 4.72 mg/g while the calculated value, *q*_e,cal_, is equal to 4.84 mg/g (*R*^2^ value of 0.9999 and *k*_2_ value of 0.151 g/(mg·min)). This comparison demonstrates that the Cu(II) adsorption process in MPP is following a pseudo-second order mechanism and its adsorption rate is controlled by the chemical adsorption of Cu(II) within the MPP. This conclusion is in agreement with that reported by Shandi et al. [21], who found that the Cu(II) biosorption by Gundelia tournefortii is more precisely described using a pseudo-second-order model than a pseudo-first-order model. Furthermore, these authors highlighted that this chemical adsorption involves complexation and ion-exchange mechanisms [21]. A comparison of the Cu(II) adsorption capacity between PP, MPP and other agri-wastes bio-sorbents reported in literature [22,23,24,25,26] is reported in Table 2. As displayed in this table, natural pomelo peels have a higher Cu(II) adsorption capacity than many natural agri-wastes bio-sorbents reported in the literature, except peach stone and pine sawdust [25]. Furthermore, the proposed chemical modification is an highly efficient method to improve performance of pomelo peels for Cu(II) adsorption. Indeed, the Cu(II) adsorption capacity of the modified pomelo peels is obviously higher than that observed with other modified bio-sorbents, except NaOH treated coconut coir, denoted CC NaOH 4.3M-RT-60, [26]. However, the Cu(II) removal efficiency (R(%) = 92.19%; i.e., 7.86 mg/g) of this alkali-treated coconut coir is lower than the R(%) value of 94.44% determined for MPP during this work.

### 2.5. Adsorption Isotherms

The impact of the initial Cu(II) concentration on the test MPP’s ability to adsorb Cu(II) was then further investigated. During this work, the adsorption isotherms were performed at initial pH = 5.0 over a range of the initial Cu(II) concentrations between 4 mg/L and 28 mg/L. The remained Cu(II) concentration in tested solution was examined by an atomic absorption spectrophotometer after a contact time of 1 h using 0.1 g of MPP and 25 mL of the tested solution. As displayed in Figure 5, the Cu(II) adsorption capacity of MPP rises from 77.3% (0.7726 mg/g) to 95.7% (6.6994 mg/g) when the initial Cu(II) concentration increases from 4 to 28 mg/L, respectively. An increase in the initial Cu(II) concentration causes a rise in the concentration gradient and the driving force, leading thus to an enhancement of Cu(II) adsorbed per weight unit of the MPP adsorbent, *q*_e_. However, at the same time, the percentage of adsorbed Cu(II) (R(%)) slightly increases over the Cu(II) concentration range in solution until reaching a plateau when the Cu(II) concentration is higher than 20 mg/L. However, at the equilibrium, the remaining Cu(II) concentration in the solution indeed also increases slightly from 0.9098 mg/L to 1.2025 mg/L by increasing the initial Cu(II) concentration from 4 mg/L to 28 mg/L. This expected drawback, associated with an increase of the Cu(II) concentration remaining in solution by increasing its initial concentration, was also observed by Yadav et al. [27] and Prakash et al. [28]. This phenomenon is simply explained thanks to a mass balance of Cu(II) in each phase as the quantity of bio-sorbent used during each experiment was kept constant. In fact, one can argue that this drawback could be also related to a limited number of active sites onto the bio-sorbent surface, being saturated with an increase in the initial Cu(II) loading concentration.

Equations related to the Langmuir (Equation (2)) and Freundlich (Equation (3)) isotherms were then used to correlate experimental adsorption data displayed in Figure 5. The Langmuir model is defined as follows:(2)$$\frac{C_{e}}{q_{e}}=\frac{1}{bQ_{max}}+\frac{C_{e}}{Q_{max}}$$
where *Q_max_* denotes the maximum monolayer capacity of the adsorbent (mg/g). *b* (L/mg) is a constant related to the heat of adsorption.

The Freundlich model is reported as expressed:(3)$$lnq_{e}=lnk_{F}+\frac{1}{n}lnC_{e}$$
where *k*_F_ (mg/g) and *n* are Freundlich isotherm constants related to the adsorption capacity and the degree of dependence of adsorption with equilibrium concentration, respectively.

The predictive capacity of both the Langmuir and Freundlich isotherm models to estimate the Cu(II) adsorption of MPP is displayed in Figure 6. As shown in Figure 6, the Freundlich model gives a better description (*R*^2^ = 0.9741) than the Langmuir model (*R*^2^ = 0.9137) for the tested Cu(II) adsorption process. This indicates that the adsorption process did not follow the Langmuir model. Therefore, the Cu(II) adsorption of MPP seems to be driven by a multilayer adsorption process.

Freundlich parameters (*k*_F_ and *n*) reveal whether the nature of adsorption is favorable. A high value of *k*_F_ suggests a significant adsorption capacity. A relatively slight slope *n* << 1 means that the adsorption intensity is favorable over the entire range of concentrations studied, while a steep slope (*n* > 1) reveals that adsorption intensity is better at high concentrations than at low concentrations. During this work, both *n* (0.157) and *k*_F_ (2.173) values of this adsorption system suggest that adsorption intensity is favorable over the entire range of concentrations studied, indicating that MPP could efficiently remove Cu(II) from tested wastewater conditions.

### 2.6. Effect of the Temperature

The effect of the temperature on the Cu(II) adsorption of MPP was then investigated at a range of 15–40 °C by keeping other experimental conditions constant (i.e., initial pH = 5.0, initial Cu(II) concentration in solution = 20 mg/L, contact time = 60 min) by putting into contact 0.1 g of MPP with 25 mL of tested solution. The gathered results show that the Cu(II) adsorption of MPP slightly increases with the temperature, even if this effect has a small impact on the MPP efficiency. In order to describe the thermodynamic behavior of the Cu(II) adsorption of MPP, thermodynamic parameters such as the change of free energy (Δ*G*^Θ^), enthalpy (Δ*H*^Θ^) and entropy (Δ*S*^Θ^) were calculated as expressed (Equations (4)–(6)):(4)$$\Delta G^{\Theta }=-RTlnK_{D}$$
where R is the gas constant *R* = 8.314 J/(mol·K); T denotes the temperature in Kelvin and *K*_D_ denotes the distribution coefficient. *K*_D_ was calculated as follows:(5)$$K_{D}=\frac{q_{e}}{C_{e}}$$
where *q*_e_ and *C*_e_ are the equilibrium Cu(II) adsorption capacity of MPP (mg/g) and the equilibrium concentration of Cu(II) in solution (mg/L), respectively.

The enthalpy (Δ*H*^Θ^) and entropy (Δ*S*^Θ^) were then estimated by using the following equation:(6)$${lnK}_{D}=\frac{∆S^{\Theta }}{R}-\frac{∆H^{\Theta }}{RT}$$

According to Equation (6), Δ*H*^Θ^ and Δ*S*^Θ^ can be calculated from the slope and the intercept when plotting ln *K*_D_ versus 1/*T*, respectively (Figure 7).

The calculated values of these thermodynamic parameters are reported in Table 3. As displayed in this table, calculated Δ*G*^Θ^ values are all negative, indicating that this adsorption process is thermodynamically feasible and spontaneous. The decrease in Δ*G*^Θ^ values with the rise in temperature reveals that the temperature has a positive impact on the Cu(II) adsorption of MPP as already depicted in Figure 7. The negative slope, highlighted in Figure 7, along with Equation (6), induces a positive Δ*H*^Θ^ value, indicating the endothermic nature of Cu(II) adsorption of MPP for the tested temperatures. The positive Δ*S*^Θ^ value suggests an increase in the randomness caused by the Cu(II) adsorption into MPP.

### 2.7. Performance of the ANN Model

In order to enlarge data points between minimum *q*_e_ and maximum *q*_e_, in present work, all experimental data of the Cu(II) adsorption capacity (*q*_e_) of MPP were then interpolated by using the cubic spline method. The cubic spline technique is easy to implement and to generate a seamless curve. The number of data points was set to 100. Then, the data were divided into training (80%) and testing (20%) sets. The number of hidden layer neurons were assessed from 4 to 12 based on the trial-and-error method. The performance of the ANN model was evaluated using the relative average absolute deviation (RAAD) and determination coefficient (*R*^2^) as expressed in Equations (7) and (8), respectively.
(7)$$RAAD(\%)=100\times \frac{\sum \left|(q_{e,exp}-q_{e,cal})/q_{e,exp}\right|}{N}$$
(8)$$R^{2}=\frac{\sum \left(q_{e,exp}-{\overset{-}{q}}_{e,exp}\right)\left(q_{e,cal}-{\overset{-}{q}}_{e,cal}\right)}{\sqrt{\sum {\left(q_{e,exp}-{\overset{-}{q}}_{e,exp}\right)}^{2}}\sqrt{\sum {\left(q_{e,cal}-{\overset{-}{q}}_{e,cal}\right)}^{2}}}$$
where *q*_e,exp_ denotes the experimental data, *q*_e,cal_ denotes the calculated data, *N* is the number of the data used, $\overset{-}{q}$_e,exp_ denotes the average value of the experimental data, $\overset{-}{q}$_e,cal_ is the average value of the calculated data.

The result of the evaluation of different neurons is shown in Table 4. Data reported in this table suggest that *R*^2^ of training reached maximum and RAAD reached minimum when the number of neurons was set equal to 9. The ANN model with a 4-9-1 (input-hidden-output) topology structure was then established as displayed in Figure 8. Figure 9 shows the training and testing performances of the developed ANN model. Training data and testing data sets were well fitted with 0.9999 and 0.9988 confidence values, respectively. Furthermore, this analysis demonstrates that the ANN model is applicable to predict the adsorption capacity of Cu(II) by MPP with a RAAD < 1.38%.

## 3. Materials and Methods

### 3.1. Chemicals

Sodium hydroxide (NaOH), nitric acid (HNO_3_), calcium chloride (CaCl_2_), ethanol and copper(II) sulfate pentahydrate (CuSO_4_·5H_2_O) used in this work were purchased from Sinopharm Chemical Reagent Co. Ltd. (Shanghai, China, analytical purity). The stock solution was initially prepared by dissolving a well-known quantity of CuSO_4_·5H_2_O in pure and deionized water, prior to be diluted to reach tested concentrations at a given pH. Each pH adjustment was done using rather HNO_3_ (0.1 mol/L) or NaOH (0.1 mol/L) solution with respect to the targeted Cu(II) concentration using the stock solution. Pomelo peels, used during this work, were obtained from a local market in Shijiazhuang City Hebei Province.

### 3.2. Material Preparation

Pomelo peels (PP) were cut into small segments, rinsed thoroughly using distilled water to eliminate dirt and dust, and then left to dry inside an oven at 80 °C for 24 h. Then, dried peel segments were crushed and sieved to remove all pieces larger than 0.5 mm for further processing. 100 g of sieved peels were steeped for 24 h in a mixture containing 250 mL of CaCl_2_ (1.5 mol/L), 250 mL of NaOH (0.5 mol/L), and 500 mL of ethanol. Peels were filtered and rinsed with deionized water until the filtrate’s pH was close to neutral. After drying at 50 °C for 24 h, the modified pomelo peels (MPP) were kept in a desiccator for further use as bio-sorbent to remove Cu(II) from wastewaters.

### 3.3. Characterization of the Tested Biosorbent

The surface features and morphology of pomelo peels were scanned with a SEM (HITACHI S-4800-I). A N_2_-adsorption technique (Micromeritics ASAP 2460), along with BET and DFT methods, was used to assess the textural properties of the tested material to determine its specific surface area and size distribution of micro- (<2 nm) and meso-pores (2–200 nm). The function groups existing in the tested material were identified by using a FTIR spectrometer (Thermo Scientific Nicolet iS20, Boston, MA, USA) with KBr discs.

### 3.4. Cu^2+^ Adsorption Tests

Batch studies were conducted in a 100 mL of stoppered conical flask, shook at 220 rpm for 1 h by using a shaking thermostat machine. The influence of the initial pH value of the Cu(II) solution on the Cu(II) biosorption using MPP was determined by combining 0.1 g of adsorbent and 25 mL of a 20 mg/L Cu(II) solution with a pH range between 2.0–7.0. Kinetic experiments were conducted as a function of the adsorption time from 0 to 60 min at (25 ± 1 °C) by keeping the initial pH constant (pH = 5.0 ± 0.2). The impact of the temperature on the Cu(II) biosorption using MPP was evaluated from 15 to 40 °C at the initial pH constant (pH = 5.0) and 1 h of contact time. During the isotherm studies, 0.1 g of adsorbent was combined with 25 mL of Cu(II) solution at various concentrations (from 4 to 28 mg/L) at the initial pH = 5.0. Samples were then withdrawn at pre-determined time intervals (from 0 to 1 h). After centrifugation at 4000 rpm for 10 min, the residual Cu(II) in wastewater was tested by using the AA-7020 atomic absorption spectrophotometer. Each experiment was conducted in duplicate. Measurements accuracy in concentrations was estimated to be close to 0.0001 mg/L. Equation (9) below was employed to calculate the quantity of Cu(II) adsorbed (*q*_e_) (mg/g):(9)$$q_{e}=\frac{(C_{0}-C_{e})V}{m}$$
where *V* represents the volume of the solution (L). *m* denotes the mass of the adsorbent (g). *C*_0_ and *C*_e_ are the initial and equilibrium Cu(II) concentrations (mg/L), respectively.

The percentage of adsorbed Cu(II) (*R*(%)) was then determined using the Equation (10):(10)$$R(\%)=\frac{\left(C_{0}-C_{e}\right)}{C_{0}}\times 100$$

### 3.5. Artificial Neural Network (ANN) Model

MATLAB 2018a was used to establish the ANN model in this work. Its topology was built by following the same methodology as that dressed experimentally. In fact, ANN inputs include experimental (1) pH; (2) contact time; (3) initial Cu(II) concentration; (4) temperature, while its output was, in fact, the Cu(II) adsorption capacity of the tested material.

Prior to perform the data training, the data were normalized to decrease the data size and accelerate training. Raw experimental data (*x*_i_) were normalized by using the following equation (Equation (11)):(11)$$x_{nor}=2\times \frac{x_{i}-x_{min}}{x_{max}-x_{min}}+(-1)$$
where *x*_nor_ is the normalized value, *x*_min_ is the minimum of the data set, *x*_max_ is the maximum of the data set, *x*_i_ is the raw experimental value that needs normalization.

Equation (12) was used to evaluate the number of hidden layer neurons by the trial-and-error method:(12)$$m=\sqrt{n+l}+\alpha$$
where *α* is a constant of value within 1 and 10, *n* denotes the number of the input layer nodes, *m* represents the number of hidden layer neurons, *l* denotes the number of output layer nodes.

During this work, the hidden and the output layers of the ANN were determined using a sigmoid-type transfer function (tansig) and a linear transfer function (purelin), respectively. Furthermore, the Levenberg-Marquardt (trainlm) technique was used to iteratively solve the proposed training algorithms, as it is a well-recognized and robust method for such a moderately sized network training [29]. The following convergence criteria were used during the ANN training: (i) maximum training times allowed was set as equal to 5000, (ii) minimum accuracy was targeted to be close to 0.0001, and (iii) a weight change rate (Mu) of 0.001 was used.

## 4. Conclusions

In this work, pomelo peels-based material was used as an agricultural waste to adsorb Cu(II) from wastewater effluents. This study shows that a rapid and simple modification of its structure could enhance its Cu(II) adsorption capacity. This improvement is mainly associated to the increase of carboxylate groups onto the modified peels surface. The highest Cu(II) adsorption capacity of MPP (4.72 mg/g) was observed by mixing 0.1 g of the MPP bio-sorbent with 25 mL of a 20 mg/L Cu(II) solution at initial pH = 5.0 and 25 °C. Effects of the initial pH, contact time, temperature and initial Cu(II) concentration on both the Cu(II) adsorption capacity and efficiency of the MPP were then investigated. Briefly, measurements show that the equilibrium is reached within 1 h and an increase of the temperature slightly impacts the Cu(II) adsorption capacity of the tested bio-sorbent. Furthermore, the Cu(II) adsorption kinetic of the MPP was found to be well fitted by the pseudo-second order equation, while the Freundlich isotherm correlated the experimental data much better than the Langmuir model. The estimated thermodynamic parameters indicate that this biosorption process is thermodynamically feasible, spontaneous, endothermic and controlled by chemical adsorption. Finally, an ANN model was developed and further evaluated during this work. The predictive capacity of the established ANN model was assessed to be close to 1.38%, indicating that this model is applicable to describe accurately the Cu(II) adsorption capacity of MPP. All the results described during this study highlight a promising feature of pomelo peels agri-wastes to be used as bio-sorbent for the removal of Cu(II) in wastewater, as an efficient green technology for ecological and environmental sustainability. The adsorption performances of the as-prepared bio-sorbent to remove selectively Cu(II) from aqueous binary and multi-component metals systems is currently investigated and will be presented and discussed in a future work.

## Figures and Tables

**Figure 1 molecules-28-03438-f001:**
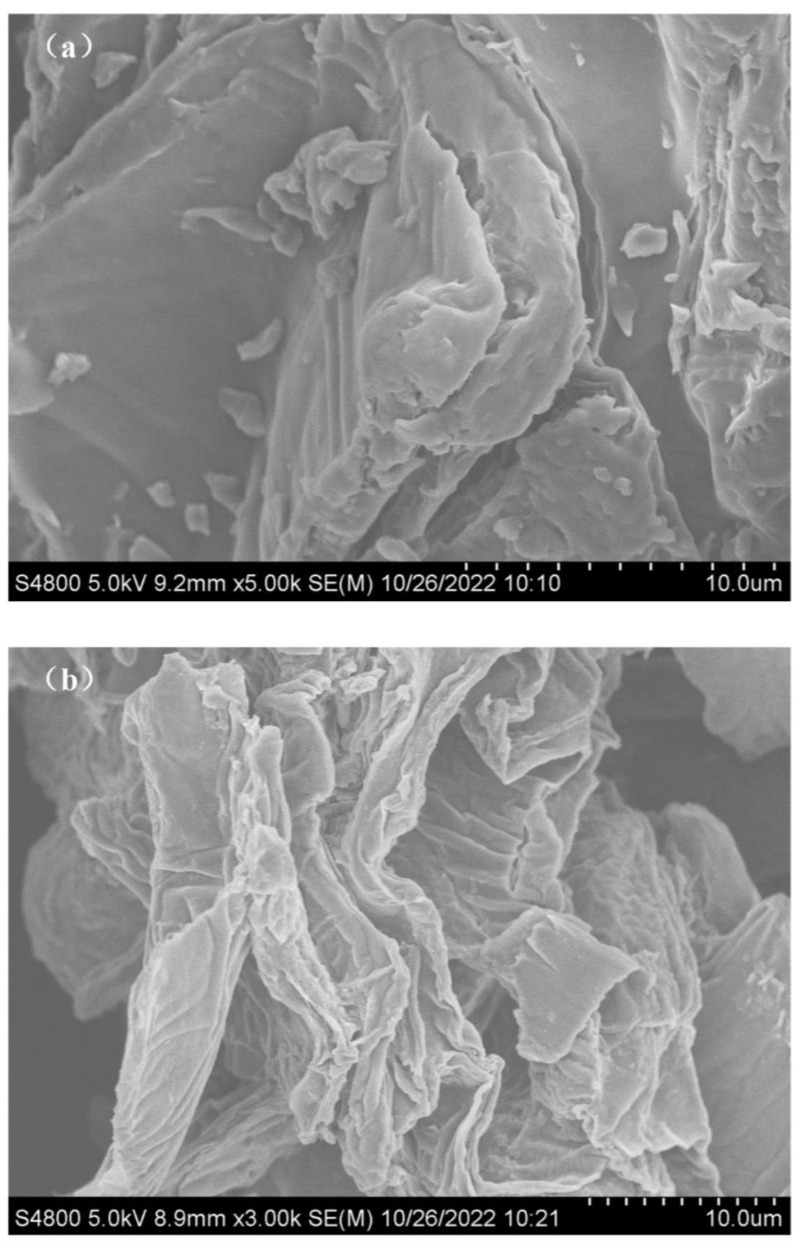
SEM images: (**a**) raw pomelo peels (PP) and (**b**) modified pomelo peels (MPP).

**Figure 2 molecules-28-03438-f002:**
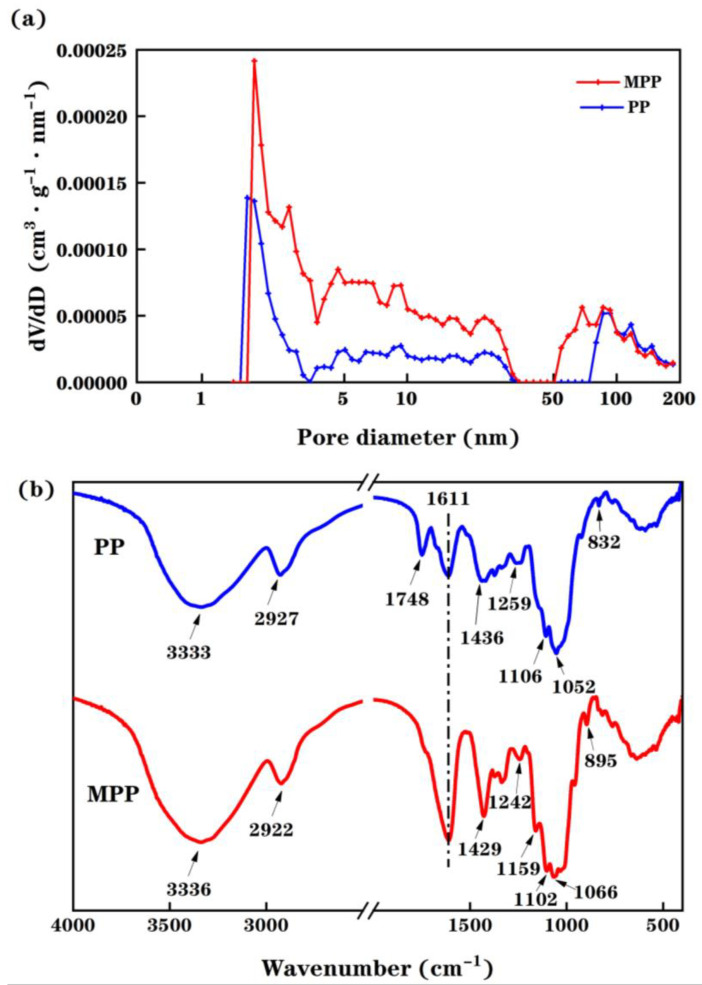
(**a**): Pore size distribution of tested adsorbents; (**b**): FTIR spectra of PP and MPP.

**Figure 3 molecules-28-03438-f003:**
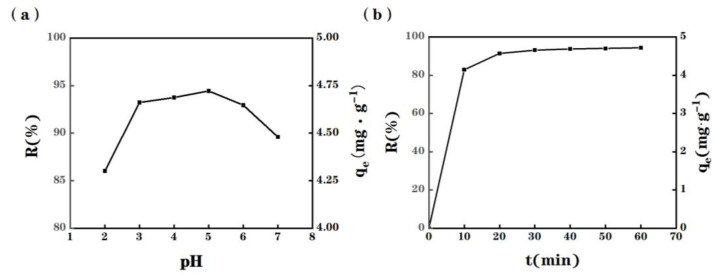
Effect of (**a**) initial pH and (**b**) contact time on Cu(II) adsorption by using 0.1 g MPP adsorbent in a 25 mL of 20 mg/L Cu(II) solution at 25 °C. Experimental data used to plot this figure are listed in the Appendix A.

**Figure 4 molecules-28-03438-f004:**
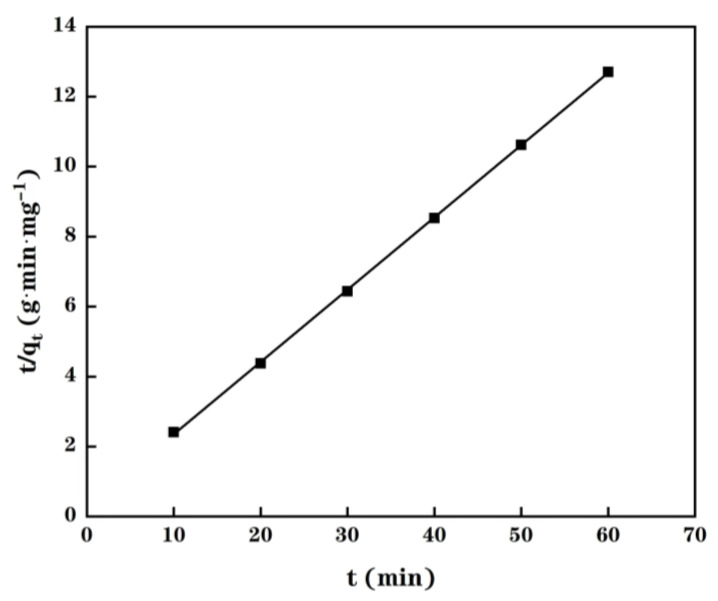
Pseudo-second-order plot of Cu(II) adsorption kinetics using data collected at 25 °C and initial pH = 5.0.

**Figure 5 molecules-28-03438-f005:**
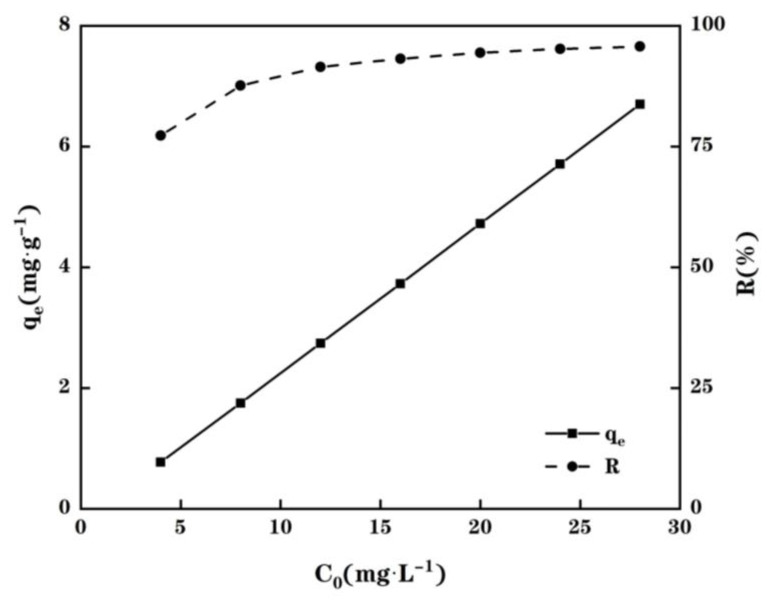
Effect of initial Cu(II) concentration on Cu(II) adsorption by using 0.1 g MPP adsorbent in a 25 mL of Cu(II) solution at a given concentration from 4 to 28 mg/L at 25 °C and initial pH = 5.0. Experimental data used to plot this figure are listed in the Appendix A.

**Figure 6 molecules-28-03438-f006:**
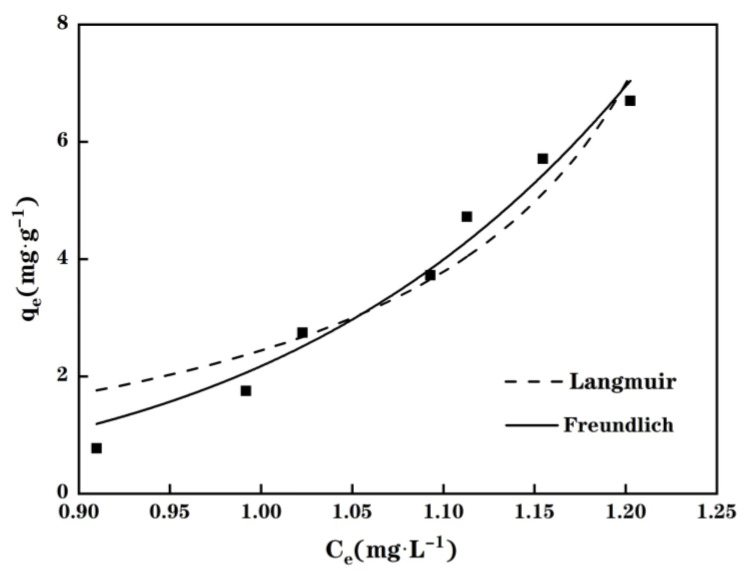
The Langmuir and Freundlich isotherms for the Cu(II) adsorption of MPP as a function of the initial Cu(II) concentration.

**Figure 7 molecules-28-03438-f007:**
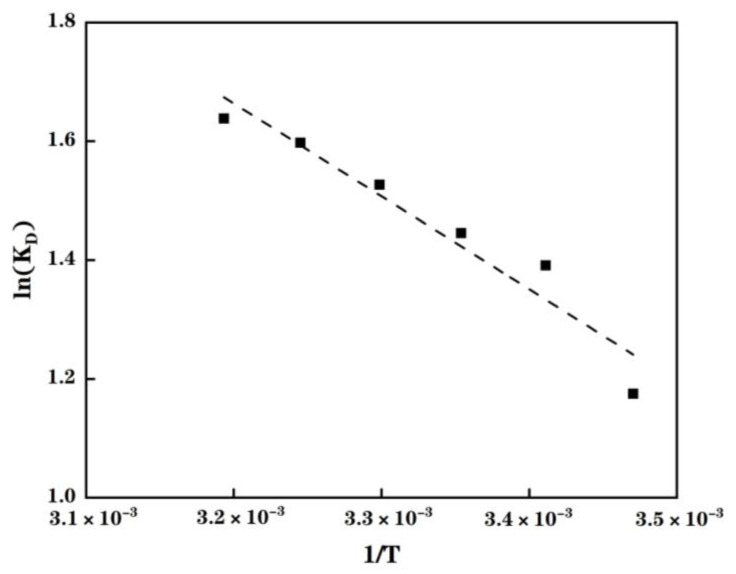
Plot of ln *K*_D_ as a function of the inverse temperature of the Cu(II) adsorption of MPP. Experimental data used to plot this figure are listed in the Appendix A.

**Figure 8 molecules-28-03438-f008:**
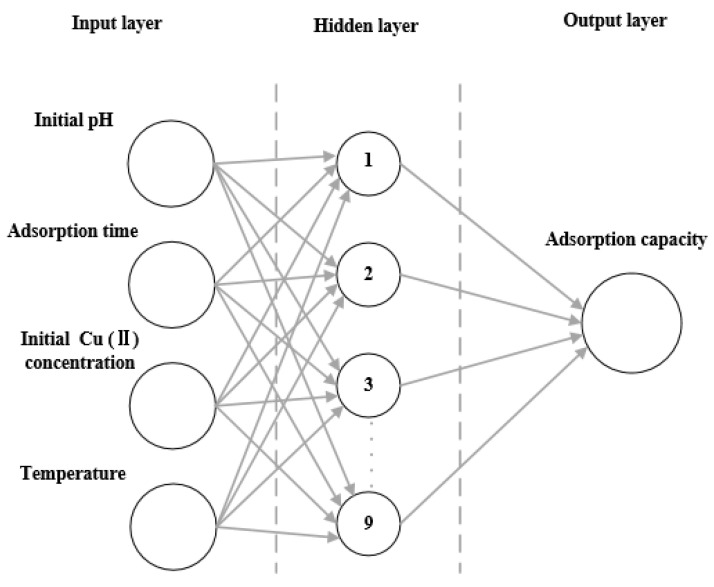
Schematic diagram of the ANN model established during this work.

**Figure 9 molecules-28-03438-f009:**
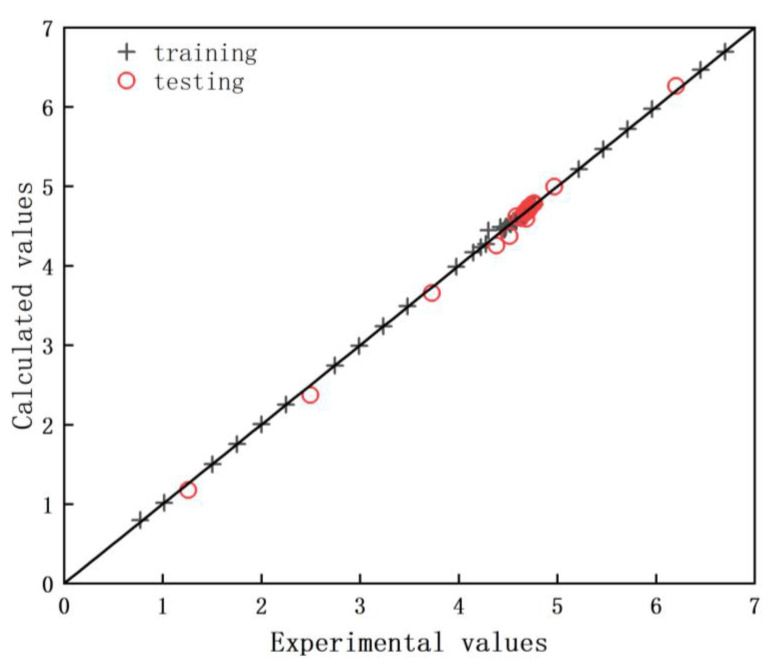
Comparison between the experimental and calculated Cu(II) adsorption capacity data for training (○) and testing (+) sets, respectively.

**Table 1 molecules-28-03438-t001:** Physical properties of PP and MPP.

	BET Surface Area (m^2^/g)	Average Pore Diameter (nm)	Pore Volume (10^−3^ cm^3^/g)
PP	0.4253	6.9269	0.737
MPP	0.9912	7.2912	1.807

**Table 2 molecules-28-03438-t002:** Comparison of the Cu(II) adsorption capacity of PP and MPP with other agri-wastes bio-sorbents reported in the literature [22,23,24,25,26].

Bio-Sorbent	Adsorption Capacity (mg/g)	Reference
Pomelo peels	3.38	This work
Modified pomelo peels	4.72	This work
Sweet sorghum residues	0.83	[22]
Fermented sweet sorghum residues	1.94	[22]
Sawdust	2.31	[23]
Natura lettuce roots	1.22	[24]
Modified lettuce roots	1.69	[24]
Natura sugarcane bagasse	0.58	[24]
Modified sugarcane bagasse	1.30	[24]
Peach stones and pine sawdust	10–15	[25]
Raw coconut coir	1.15	[26]
CC NaOH 4.3M-RT-60	7.86	[26]

**Table 3 molecules-28-03438-t003:** Thermodynamic parameters for the Cu(II) adsorption by MPP.

T (K)	*K* _D_	Δ*G*^Θ^ (KJ/mol)	Δ*H*^Θ^ (KJ/mol)	Δ*S*^Θ^ (J/(mol·K)
288.15	3.2377	−2.8146	13.01	55.46
293.15	4.0193	−3.3905		
298.15	4.2428	−3.5825		
303.15	4.6039	−3.8484		
308.15	4.9403	−4.0925		
313.15	5.1455	−4.2649		

**Table 4 molecules-28-03438-t004:** Evaluation results for the number of neurons in the hidden layer.

m	4	5	6	7	8	9	10	11	12
RAAD (%)	0.255	0.644	0.621	1.183	0.713	0.038	0.324	0.533	0.417
*R* ^2^	0.9998	0.9997	0.9997	0.9991	0.9978	0.9999	0.9999	0.9999	0.9989

## Data Availability

Data is contained within the article or the Appendix A.

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
