# Peer review of "Assessment of Cu(II) Removal from Aqueous Solutions by Modified Pomelo Peels: Experiments and Modelling"

_molecules, 2023, doi:10.3390/molecules28083438_

Round 1
Reviewer 1 Report
The manuscript molecules-2337873 "Assessment of Cu(II) Removal from Aqueous Solutions by Modified Pomelo Peels: Experiments and Modelling" presents interesting results about wastewater treatment. The paper presents relevant information. It is well designed and easy to follow. The applied methodology is adequate and the literature cited is highly actualized. It deserves publication in Molecules after Major Revisions. The authors must consider the following comments:
- Provide the meaning of the acronyms SEM, FTIR, and BET in the abstract.
- Change the keywords that are in the title. Provide more innovative keywords.
- The abstract must be rewrite. It should contain a brief introduction, the methodology, the main results and end with a conclusion about importance of the article.
- Revise line 72, reference number 176 does not exist.
- The end of the introduction section should be rewritten. The authors must justify the reason for the study and what the objectives were.
- Materials and Methods must appear after the Introduction section.
- The font of the x and y axes of figure 3 need to be enlarged. Redo it.
- Copper removal kinetics can be compared with other studies. Add them according to context.
https://doi.org/10.1080/09593330.2020.1793005
https://doi.org/10.1021/ie9903137
- English needs to be improved throughout the manuscript. Avoid using long sentences and the overused of “which”. There are 13 “which” throughout the manuscript.
- The abbreviation of minutes is min. Revise it.
- The article presents only 20 references, of which 14 are in the Introduction.
The authors need to better endorse the discussion of the results using the literature. Add other studies that support the results found.
- The conclusion section should be rewritten. A conclusion brings what the authors conclude with the results of the article. What advance did the article bring to this area?
Reviewer 2 Report
This manuscript was well presented and should be published after revision. I only have one concern about the ANN model, as there are a variety of factors that could affect the sorption process, such as sorbent dosage, co-existing ions, ionic strengths, etc. I recommend the authors to add this information in the section of Implications to make the model more reliable in the future.
Reviewer 3 Report
Recently, environmental issues have come to the fore. The reviewed manuscript discusses some physico-chemical properties of an adsorbent based on pomelo peel. The authors focused their attention on the issues of adsorption of copper (ii) ions from an aqueous solution. The limits of the effective operation of the new adsorbent in terms of temperature, copper concentration and acidity of the solution have been established. The direction proposed by the authors of the manuscript will allow not only to remove toxic copper ions from the water, but also to dispose of food industry waste – pomelo peel. Despite the positive aspects of this manuscript, the reviewer has a number of questions to its authors:
1. What determines the choice of the object of research? Is it also possible to use the peel of other citrus fruits – oranges, grapefruit, etc.?
2. Did the authors compare the effectiveness of the new adsorbent with the effectiveness of an adsorbent based on a mixture of cellulose and lignin in the same proportion as in the pomelo peel?
3. In my opinion, Figure 3 should be divided into several figures or moved to the end of section 2.6.
4. I would like to draw the attention of the authors of the manuscript to the fact that despite the increase in the proportion of adsorbed ions with an increase in the concentration of copper in the solution (lines 196 and 197), the amount of remaining copper in the water increases (!). According to the research results, after treatment with an absorbent 1 liter of solution with a concentration of 4 mg / L in water, 0.908 mg of copper will remain in water, while after treatment 1 liter of solution with a concentration of 28 mg / L 1.204 mg of copper remains in water.
5. It is necessary to correct the links. The names and surnames of the authors are mixed up in them.
Round 2
Reviewer 1 Report
The authors performed allsuggested corrections. Therefore, the paper must be accepted.
Reviewer 3 Report
The revised version of the manuscript contains all the necessary information and can be published after editorial revision